# HOCl Responsive Lanthanide Complexes Using Hydroquinone Caging Units

**DOI:** 10.3390/molecules25081959

**Published:** 2020-04-23

**Authors:** Elena Del Giorgio, Thomas Just Sørensen

**Affiliations:** Nano-Science Center & Department of Chemistry, University of Copenhagen, Universitetsparken5, 2100 København Ø, Denmark; EXD942@student.bham.ac.uk

**Keywords:** lanthanide coordination chemistry, lanthanide luminescence, responsive molecular probes, ROS probes, reactive oxygen species

## Abstract

Redox biology is still looking for tools to monitor redox potential in cellular biology and, despite a large and sustained effort, reliable molecular probes have yet to emerge. In contrast, molecular probes for reactive oxygen and nitrogen have been widely explored. In this manuscript, three kinetically inert lanthanide complexes that selectively react with hypochlorous acid are prepared and characterized. The design is based on 1,4,7,10-tetraazacyclododecane-1,4,7-triacetic acid (DO3A) and 1,4,7,10-tetraazacyclododecane-1,7-diacetic acid (DO2A) ligands appended with one or two redox active hydroquinone derived arms, thereby forming octadentate ligands ideally suited to complex trivalent lanthanide ions. The three complexes are found to react selectively with hypochlorous acid to form highly symmetric lanthanide(III) 1,4,7,10-tetraazacyclododecane-1,4,7,10-tetraacedic acid (DOTA) complexes. The conversion of the probe to [Ln.DOTA]^−^ is followed by luminescence, absorption, and NMR spectroscopy in a model system comprised of a Triton-X modified HEPES buffer. It was concluded that the design principle works, and that simple caging units like hydroquinones can work well in conjugation with lanthanide(III) complexes.

## 1. Introduction

Reactive Oxygen Species (ROS), and Reactive Nitrogen Species (RNS) [1,2,3], are classes of compounds that include one- or two-electrons oxidants and play essential roles in living organisms [1,2]. The most relevant ROS originate from a cascade started by the one electron reduction of O_2_ giving the superoxide radical (O_2_**^−^)** by means of either NADPH oxidases (NOXs) or by leakage in the electron transport chain inside the mitochondria [1,3,4,5,6,7,8]. Superoxide Dismutase (SOD) is responsible for the conversion of superoxide into hydrogen peroxide (H_2_O_2_) which, in turn, can either undergo a Fenton reaction in the presence of Fe^2+^ (or Cu^2+^) to give the hydroxyl radical OH**^·^** or convert to hypochlorous acid by action of Myeloperoxidase (MPO). ROS play numerous and diverse roles in biology: Biomolecules are responsive to redox conditions and many biological processes are regulated by redox chemistry [1,2,6,7]; although the first understanding of ROS concerned either their deleterious action against lipids, proteins and nucleic acids or their role as antimicrobial agents, the importance of these reactive molecules has recently been revised, spurring the need for better probes to detect them [4,9,10]. Given the key role of ROS in biology, many methods have been developed in order to detect them. Among these, mass spectrometry and a plethora of fluorescent and luminescent probes have been proposed over the years [4,11,12,13,14]; these probes have also been conjugated to units (peptides or TTP) that can target specific organelles [4]; however, because of the complexity of both living organisms and redox processes in vivo, the selective and reliable detection of ROS has proved tricky [9,15,16]. Here, we consider luminescent probes for ROS detection that are based on lanthanide(III) luminescence [9,17].

Lanthanide ions possess unique luminescence properties [18]: Lanthanide(III) ions gives rise to high sensitivity even at low concentrations [19], have sharp and characteristic emission bands, and long luminescent lifetimes in the order of µs-ms [18,20]. The excited states of lanthanide(III) ions are not quenched by O_2_ and the distinctive and sharp bands boost both signal to noise ratio and sensitivity to analytes. Additionally, DOTA-like lanthanide(III) complexes are fully water soluble, non-toxic and inert in physiological conditions [21]. In a typical probe design, the Ln^3+^ ion is coordinated to a chelator to avoid release of the toxic free ion in the media and to confer the desired features [18,22,23]. Macrocycles like DOTA and its derivatives are more rigid and can form complexes with higher kinetic inertness than open chain chelators [22,24]. Thanks to the kinetic inertness, these complexes can be seen as molecular building blocks and can be linked to different units [23]. The sensing of an analyte can arise from either direct coordination of the analyte to the lanthanide ion centre or from binding/reaction of the analyte with a responsive unit on the chelator. Lanthanide-based probes have been developed to detect an array of analytes [25], e.g., metal ions [26,27,28], anions [29], ATP [30,31,32,33,34], ROS [13,35,36], and pH [37,38].

The majority of molecular probes for Reactive Oxygen and Nitrogen Species (RONS) are based on organic dyes, although some examples of lanthanide(III)-based probes have been reported [11,39,40,41,42,43]. Most probes are based on a cage on/off mechanisms: the cage acts as a leaving group upon reaction with the RONS, and in the lanthanide(III)-based probes, the cage blocks the energy transfer from an antenna to the lanthanide centre. When the cage reacts with the RONS, it is cleaved and the antenna is free to transfer energy to the lanthanide(III) centre [44]. Examples of caging units are pinacol boronic esters [36,45], p-aminophenoxy groups [13,46], and dinitrophenyl groups [47], and probes based on these are reported as selective for H_2_O_2_ and hROS (Highly Reactive Oxygen Species, HOCl and **^·^**OH). Borbas et al. developed a versatile platform built around the deprotection of a coumarin, thus unlocking its sensitization activity towards Eu^3+^ [45]. Another approach to the design of responsive complexes is the in situ formation of an antenna upon reaction with the ROS. Pierre et al. developed a library of pre-antennas based on aromatic acids or amides; these are hydroxilated in the presence of **^·^**OH and coordinate to a Tb.DO3A complex, thus enhancing the luminescence intensity [48]. Finally, a ferrocene appended Eu.DO3A complex prepared by Faulkner should be mentioned. This probe reversibly responds to redox processes [49]. The relevance of this probe is that, unlike the ones reported here, it does not depend on an irreversible process. The advantage of a reversible detection is that saturation of the probe is avoided and a continuous monitoring of the system’s redox state would be feasible [9,50,51].

In this work, we also adopt a design based on a cage, although exploiting a different mode of action. The complexes shown in Figure 1 all carry a hydroquinone cage that is released upon reaction with hypochlorous acid forming quinone and the same fully symmetrical [Ln.DOTA]^−^ complex, see Scheme 1, a transformation that can be followed using luminescence, absorption, and NMR spectroscopy. The results presented here validate the design concept, and further show that the incorporation of the caging group in a lanthanide(III) complex significantly reduces the reactivity. Finally, it should be noted that the complexes explored here, does not incorporate an antenna chromophore. While this does not constitute an issue in our experiments, it limits the applicability of the complexes studied as molecular probes.

## 2. Results and Discussion

We designed and synthesized three different ligands and prepared their europium(III) and terbium(III) complexes. Considering the three lanthanide(III) complexes Ln.**L^1^** and Ln.**L^2^** are inherently different from Ln.**L^3^** as the former two are based on the 1,4,7,10-tetraazacyclododecane-1,4,7-triacetic acid (DO3A) scaffold, while the latter is a 1,4,7,10-tetraazacyclododecane-1,7-diacetic acid (DO2A) derivative. The symmetry of the complexes is different, which is expected to be reflected in both NMR and luminescence spectra. Finally, neither of the complexes contains a chromophore, which infers that the relatively inefficient direct excitation into the lanthanide(III) excited state manifold is required if these complexes are to be used as molecular probes.

Considering the three complexes as reactive probes for RONS, there are clear differences that are probed by using these structures. Where Ln.**L^1^** and Ln.**L^3^** both carry mono-esters of hydroquinone, one and two respectively, Ln.**L^2^** has two lanthanide binding pockets on a di-ester of hydroquinone. Please note that both Ln.**L^2^** and Ln.**L^3^** are converted to Ln**.L^1^** upon the first reaction with RONS, see Scheme 1. Thus, the reactivity of the two hydroquinone esters can be compared, and it can be concluded if the incorporation of subsequent un-caging is a viable route towards higher selectivity. To perform these comparisons, a reliable testbed for the reaction between RONS and the lanthanide(III) complexes was required.

### 2.1. Oxidation Testbed

To investigate the reaction between RONS and various probes, we tested several solvent and buffer systems. We tested using hydroquinone and the naturally occurring antioxidant α-tocopherol in order to simulate the different solubilities and reactivities of various probes. We selected to work at neutral pH as the end target is probes for molecular biology. Figure 2 shows the results obtained in our optimized buffer system which consists of 100 mM HEPES buffer with 1–3% Triton X-100 depending on the lipophilicity of the probes investigated. These conditions provide beautiful absorption spectra, where the oxidation of hydroquinone and α-tocopherol can be followed. Please note that full conversion is achieved when 1.5–2.0 equivalents of hypochlorous acid have been added.

### 2.2. Synthesis

The target molecules, shown in Figure 1, are made from the four building blocks shown in Figure 3. The pendant arms **1** and **3** are obtained via nucleophilic substitution reactions between chloroacetyl chloride and hydroquinone in high yields as white crystals and ice white flakes, respectively (Scheme 2). Protection of the phenol unit in **1** is carried out with *tert*-butyldimethylsilyl chloride and *N*-methyl imidazole in DMF to give the protected pendant arm **2** as a colourless oil in 64% yield after purification.

Protected ligands **4** and **5** are synthesized by linking the pendant arms **2** and **3**, respectively, to the DO3A-triester via nucleophilic substitution. For **4**, stirring in MeCN with NaHCO_3_ for 26 h gives the desired compound as a transparent oil in moderate yields (55–60% over 4 runs of the reaction). Compound **5** is obtained in 52% yield as a yellow viscous oil in similar conditions under reflux and with DIPEA as base, see Scheme 3.

For the preparation of protected ligand **9**, the reported procedure [52], as seen in Scheme 4, first requires the formation of the regioselectively disubstituted cyclen with the carboxylbenzyl group. The synthesis is carried out with benzyl chloroformate at 60 °C in chloroform for 100 min, giving the disubstituted compound as a colourless oil in 57% yield after work up. The introduction of *tert*-butyl acetate groups is conducted in refluxing acetonitrile with DIPEA for 30 min in excellent yield, giving **7** as a pale yellow oil. The removal of the carboxylbenzyl groups is achieved by microwave irradiation in isopropanol catalyzed by Pd/C with ammonium formate as source of hydrogen, yielding compound **8** as a white solid (98%). The resulting DO2A-diester is reacted with the pendant arm **2** and DIPEA in refluxing acetonitrile for 2 h, giving the protected ligand **9** as a pale yellow oil in reasonable yields after purification by column chromatography (50–55% after 2 runs of the reaction).

The deprotection of both the *tert*-butyldimethylsilyl and *tert*-butyl groups is obtained by stirring the protected ligand with TFA in DCM at room temperature. The desired compound is retrieved after multiple triturations from MeOH and diethyl ether to remove excess TFA. Scheme 5 shows the reaction for the formation of ligand **L^1^**, obtained as a white solid in 39% yield. **L^2^** is attained in the same conditions and in comparable yields to **L^1^** as an off white solid (32%). In the case of **L^3^**, 72 h are required to get acceptable yields (36%) and give the desired ligand as an off-white powder.

The lanthanide (III) complexes of Eu (III) and Tb (III) with **L^1^**, **L^2^** and **L^3^** are obtained as white powders by reacting the ligand with either 1 equivalent (**L^1^** and **L^3^**) or 2 equivalents (**L^2^**) of the lanthanide (III) triflate salt in MeOH at 60 °C for 72 h.

### 2.3. Characterisation

The synthesized ligands and complexes were characterized using mass spectrometry and NMR. The ^1^H NMR spectra of the three europium(III) complexes Eu.**L^1^**, Eu.**L^2^** and Eu**.L^3^** are shown in Figure 4 and contrasted to the spectrum of [Eu.DOTA]^−^. Eu.**L^1^** has no symmetry and resonances from the four axial protons are observed, whereas only a single resonance is seen in the DOTA complex. Eu.**L^2^** and Eu**.L^3^** are both C2 symmetric and a much simpler ^1^H NMR spectrum is observed. Eu**.L^3^** shows a spectrum very similar to the C4 symmetric DOTA complex. So does Eu.**L^2^**, but here the symmetry is broken in a fraction of the population, possibly by association to a capping ligand [53]. Please note that all complexes in Figure 4 are predominately in the SAP conformation.

The ^1^H NMR spectra indicate a DOTA like structure of all complexes, with added conformational freedom in Eu**.L^2^**. This is confirmed by the luminescence emission spectra, where the emission spectrum of Eu**.L^2^** is more similar to that of DO3A than DOTA [53]. In contrast, the emission spectra of Eu**.L^1^** and Eu**.L^3^** are both structured, and the band at 590 nm corresponding to the ^5^D_0_→^7^F_1_ transition shows the different symmetry of the two complexes. Three lines are resolved for the asymmetric Eu**.L^1^**, and two lines are seen in the symmetric Eu**.L^3^**. The luminescence spectra are included in Figure 5. Time-gating did not provide any benefits in the model system and only steady-state spectra are used in this study.

The last panels in Figure 5 show the steady-state and time-gated emission spectrum of Tb.**L^1^**. Excitation in the absorption band of the hydroquinone unit clearly gives rise to terbium(III) centred emission, with appreciable efficiency. Thus, the terbium(III) complexes can be used as intensity-based probes, as oxidation will lead to loss of the hydroquinone unit which is followed by a significant loss of emission intensity. The caveat for using this mode of operation is that it requires excitation in the deep UV. The excitation spectrum in Figure 5b shows that direct excitation of europium(III) is more efficient than the antenna-mediated excitation, and this will not be considered further. All excitation and emission spectra are available as Appendix A.

The luminescent lifetimes of the complexes were determined using an exponential fit to the time-resolved emission decay profiles. The results are compiled in Table 1, and show clear differences between the complexes. In Tb.**L^2^** and Tb.**L^3^** the terbium excited state lifetime is reduced by ligand specific quenching pathways that are absent in Tb.**L^1^**. Similarly, the luminescent lifetime of Eu**.L^1^** also resembles those observed from other DO3A complexes [53], while those observed in Eu**.L^2^** and Eu**.L^3^** are significantly reduced. This suggest additional quenching pathways operate in Ln.**L**^2^ and Ln.**L^3^**, both redox mediated charge transfer and back energy transfer to the antenna centred triplet state are viable explanations. As the luminescence intensity is important for an efficient molecular probe, further developments should exploit the Ln.**L^1^** scaffold as—based on the luminescence lifetime—this is the most promising candidate of the three.

### 2.4. Response to Reactive Oxygen Species

The response of Eu**.L^1^**, Eu**.L^2^**, and Eu**.L^3^** towards ROS were studied in detail, and the response of Tb.**L^1^**, Tb.**L^2^**, and Tb.**L^3^** when exposed to hypochlorous acid was investigated. The complexes were dissolved in 0.1 mM HEPES buffer without Triton X-100 as all complexes are fully soluble. H_2_O_2_ or HClO were added and after 5 min the spectra were recorded. No reaction was observed with hydrogen peroxide, and we concluded that the complexes are selective for hypochlorous acid.

The response of all six complexes are due to the oxidation of hydroquinone to quinone, and a subsequent hydrolysis of the Ln.DOTA-ester to [Ln.DOTA]^−^. The transformation is shown for Eu.L_1_ in Scheme 1. For Ln**.L^2^** and Ln.**L_3_** the first oxidation/hydrolysis step leads to the formation of Ln.**L_1_**, which then upon a second oxidation is converted to [Ln.DOTA]^−^. Figure 6 shows the spectra of Eu**.L^1^** when 1–25 equivalents of HOCl have been added. There is a small observable change in intensity, but the most significant change is in the band at 590 nm arising from the ^5^D_0_→^7^F_1_ transition. In this band-consisting of three lines-the ligand field can be probed via the three microstates in ^7^F_1_. In Eu**.L^1^**, a DO3A system, the three microstates are all resolved, while in the product there are two degenerate microstates. The product spectrum is very similar to that of [Eu.DOTA]^−^, and that the product is indeed [Eu.DOTA]^−^, can be seen from the series of ^1^H NMR spectra shown in Figure 7. This is further supported by the direct comparison of the Eu**.L^1^** oxidation product luminescence emission spectrum to that of [Eu.DOTA]^−^, see Figure 8a. Thus, the mechanism of the response is confirmed.

The response of the probes can be determined in several ways. Figure 6 shows the ratiometric change in the emission spectrum as the symmetry of the complexes change, which can be converted into a response function by comparing the intensity at two emission wavelengths as done in Figure 8b. Figure 8c shows how the intensity of antenna mediated europium(III) luminescence decreases as the caging group is hydrolysed, and Figure 8d shows the corresponding response function. Finally, Figure 8e,f shows how the overall intensity increase of the Ln.**L^3^** luminescence can be used as a response. The change in spectral shape can be used for europium(III) only. The intensity response works equally well for europium(III) and terbium(III) complexes.

Figure 8 includes the response function of the three probes as a function of equivalents of HOCl added. Compared to the response function of the free hydroquinone, a significant excess of oxidant is needed to achieve full conversion of the probes. For all three complexes an excess of 15 equivalents of HOCl is needed, which indicates that the lanthanide complex significantly reduces the reactivity of the hydroquinone ester. Please note that the mono-esters of hydroquinone in Ln**.L^1^** and Ln**.L^3^** show similar reactivity as the di-ester of hydroquinone in Ln**.L^2^**, although full conversion is reached at 30 equivalents for Ln**.L^2^** and 15–25 equivalents with the mono-esters. Furthermore, the two reactions that are required for full conversion of Ln**.L^2^** and Ln**.L^3^** are not reflected in the number of equivalents needed for full conversion. Thus, we conclude that the reactivity of the hydroquinone esters is dictated by the structure of the lanthanide complex, rather than the hydroquinone esters themselves. This is supported by the fact that the reaction of the model systems, see Figure 2, proceed with full conversion by addition of the expected two equivalents of hROS.

Although the reported probes have been tested for HOCl detection, it is informative to examine its response towards other ROS. Eu.**L^1^** has been studied with H_2_O_2_ and the H_2_O_2_/Horseradish Peroxidase system in PBS buffer with Triton X-100. The results are reported in Figure 9a,b, respectively. The emission spectra of the complex are unchanged after addition of H_2_O_2_ alone or in combination to Horseradish Peroxidase, thus supporting the aim of developing a selective probe towards a single ROS.

## 3. Materials and Methods

### 3.1. General

All solvents and chemicals were purchased from Sigma-Aldrich, TCI chemicals, Acros Organics, Alfa Aesar, Fluka, Merck, Abcr or VWR chemicals. The solvents were HPLC or technical grade. Demineralized H_2_O was used. THF was distilled over Na/benzophenone. DMF, DCM and Toluene were dried over activated molecular sieves (4A). Inert atmosphere for reactions was obtained by either maintaining a constant argon flow from a balloon filled with argon and connected through a syringe needle in a septum or from constant flow of nitrogen through a bubbler. All synthesized compounds containing air-sensitive units were stored under inert gas in small vials for a maximum of 5 weeks and, after that time, their condition was checked before use. Column chromatography was performed on silica gel pore size 60 Å, particle size 40–63 μm, 239–400 mesh from Rocc. Silica gel plates 60 F254 from Merck were used for thin layer chromatography. ^1^H-NMR and ^13^C-NMR spectra were recorded on a Bruker 500 MHz and 126 MHz respectively. All chemical shifts (δ) are expressed in parts per million (ppm). The deuterated solvents were used as internal references with the reported values (CDCl_3_ δ*H* 7.26 δ*C* 77.16, CD_3_OD δ*H* 3.31 δ*C* 49.00, D_2_O δ*H* 4.79). ESI^+^-MS spectra were recorded on a Bruker Solarix FTICR instrument. MALDI-TOF spectra were recorded on a Bruker Speed Autoflex with dithranol as matrix. GC-MS traces and spectra were recorded on an Agilent 6890 Series (GC system) and Agilent 5973 Mass Selective Detector with *tert*-butylmethyl ether as solvent for injections. Melting points are uncorrected and measured on a Stuart SMP30 instrument. All aqueous solutions were prepared in demineralized water. The pH for buffer solutions was measured with a Mettler-Toledo Seven Easy pH meter. HEPES buffer 0.1 M was prepared from 2.383 g of purchased HEPES in 100 mL of demineralized H_2_O. PBS buffer 20 and 40 mM was prepared from diluting purchased PBS buffer 100 mM. Aqueous solution of HOCl 5% *w*/*v* was prepared from 384 mg of purchased Ca(OCl)_2_ (65% available chlorine) in 5 mL of demineralized H_2_O. Deuterated DOCl 5% *w*/*v* solution was prepared from 38.4 mg Ca(OCl)_2_, 65% available chlorine, in 0.5 mL D_2_O. H_2_O_2_ 30% *w*/*v* was used as purchased. H_2_O_2_ 3% *w*/*v* was prepared from dilution of the purchased H_2_O_2_ 30% *w*/*v*. A solution of Horseradish Peroxidase 750 U/mL was prepared from 1.41 mg of Horseradish Peroxidase (E.C. 1.11.1.7, 150 U/mg) in 0.282 mL PBS 40 mM at pH 6.5.

Absorption spectra were recorded on a Cary 300 UV-Vis Spectrophotometer at room temperature and using pure solvent as baseline. The slits were set at 2 nm, the integration time was 0.1 s and the step size was 1 nm. Excitation spectra were acquired on a HORIBA PTI QuantaMaster 8075–22 instrument with a Steady State Xenon Arc lamp. The excitation slits were set at 2 nm and the emission slits at 7 nm, the integration time was 0.15 s and the step 1 nm for all experiments unless stated otherwise. The spectra were recorded at 25 °C. The excitation and emission wavelengths are reported on the spectra. The uncorrected spectra were used for qualitative individuation of peaks. Emission spectra were acquired on a HORIBA PTI Quanta Master 8075–22 instrument with a Steady State Xenon Arc lamp. The excitation slits were set at 8 nm and the emission slits at 2 nm, the integration time was 0.1 s and the step 1 nm for all experiments unless stated otherwise. The spectra were recorded at 25 °C. The excitation and emission wavelengths are reported on the spectra. The reported spectra were corrected with the correction file provided with the software. Time-gated emission spectra were acquired on a Cary Eclipse fluorescence spectrometer in phosphorescence mode. The delay time was 0.15 ms with a gate time of 5 ms. The excitation slits were set to 10 nm and the emission slits to 5 nm or 2.5 nm. Lifetime measurements were obtained on a HORIBA PTI QuantaMaster 8075–22 instrument with a Flash Xenon lamp. The excitation slits were set at 8 nm and the emission slits at 8 nm, the start time was 300 μs and end time 10,000 μs. The spectra were recorded at 25 °C. The excitation and emission wavelengths are reported on the spectra. The data was fitted using OriginPro 2017.Starna Scientific 10 mm quartz cuvettes were employed for all samples at room temperature.

### 3.2. Synthetic Procedures

#### 3.2.1. Synthesis of 4-Hydroxyphenyl 2-chloroacetate (**1**)

Hydroquinone (1.00 g, 9.08 mmol) was dissolved in DMF (18.1 mL). Chloroacetyl chloride (0.866 mL, 10.9 mmol) was added dropwise under a positive nitrogen atmosphere over 10 min and the clear solution soon became pale yellow. After 45 min the reaction was quenched by addition of 20 mL H_2_O and the white precipitated diester was filtered off. The filtrate was extracted with Et_2_O (20 mL × 3). The combined organic layers were washed with H_2_O (30 mL × 2) and brine, dried over MgSO_4_ and concentrated in vacuo to yield the crude compound **1** as a white powder. The crude was recrystallized in H_2_O and pure compound **1** was obtained as white crystals (1.44 g, 7.72 mmol, 85%). ^1^H-NMR (500 MHz, CDCl_3_): δ 6.99 (d, *H_ar_*, 2H), 6.84 (d, *H_ar_*, 2H), 4.28 (s, Cl-*CH_2_*-CO, 2H) ^13^C-NMR (300 MHz, CDCl_3_): δ 153.75 (C4) 122.29 (C_ar_), 116.21 (C_ar_), 41.01 (Cl-*C*-CO) ESI^+^-MS: C_8_H_7_ClO_3_ [M + Na]^+^
*m*/*z*_calc_ 208.99759 *m*/*z*_found_ 208.99778 λ_abs_ (HEPES 0.1 M, pH 7.2) = 275 nm, 221 nm.

#### 3.2.2. Synthesis of 4-((Tert-butyldimethylsilyl)oxy)phenyl 2-chloroacetate (**2**)

TBDMSCl (400 mg, 2.14 mmol) was dissolved in DMF (4.2 mL) and *N*-Methyl Imidazole (0.64 mL, 8.04 mmol) was slowly added under a positive nitrogen atmosphere. The mixture was allowed to stir for a few minutes and then a solution of **1** (500 mg, 2.68 mmol) in DMF (1.9 mL) was introduced. The solution was stirred at room temperature for 4 h and then quenched by addition of H_2_O (10 mL). The mixture was extracted with Et_2_O (15 mL × 3) and the combined organic layers were washed with H_2_O (20 mL × 3) and brine. After drying over MgSO_4,_ the solvent was evaporated under reduced pressure to yield a colourless oil. The crude product was purified by column chromatography (8:2 Petroleum Ether: EtOAc) and dried for a day on the oil pump to yield the pure compound **2** as a colourless oil (516 mg, 1.72 mmol, 64%). ^1^H-NMR (500 MHz, CDCl_3_): δ 6.99 (d, *H_ar_*, 2H), 6.83 (d, *H_ar_*, 2H), 4.27 (s, Cl-*CH_2_*-CO, 2H), 0.98 (s, Si-C-*CH_3_*, 9H), 0.19 (s, Si-*CH_3_*, 6H) ^13^C-NMR (300 MHz, CDCl_3_): δ 166.21 (CO), 153.85 (C4), 121.95 (C_ar_), 120.79 (C_ar_), 41.05 (Cl-*C*-CO), 25.78 (Si-C-*(CH_3_)_3_*), 18.32 (Si-*C*-(CH_3_)_3_, −4.32 (Si-CH_3_) ESI^+^-MS: C_14_H_21_ClO_3_Si [M + H]^+^
*m*/*z*_calc_ 301.10213 *m*/*z*_found_ 301.10223.

#### 3.2.3. Synthesis of 1,4-Phenylene bis(2-chloroacetate) (**3**)

Hydroquinone (1.00 g, 9.08 mmol) and imidazole (1.36 g, 19.98 mmol) were dissolved in DMF (18.2 mL). Chloroacetyl chloride (1.59 mL, 19.98 mmol) was added dropwise over 5 min under a positive argon atmosphere and the mixture was left stirring overnight at room temperature. The colour of the solution changed from colourless to bright yellow. H_2_O (25 mL) was added to quench the reaction and a part of compound **3** precipitated as a white powder. This was filtered off and repeatedly washed with H_2_O. The filtrate was then extracted with DCM (15 mL × 3), the organic layers were washed with H_2_O (15 mL × 2) and brine. After drying over MgSO_4_, the solvent was evaporated to yield a white powder. The residue was dissolved in the minimum amount of EtOH and triturated with H_2_O to yield pure compound **3** (2.21 g, 8.36 mmol, 92%) as ice white flakes. ^1^H-NMR (500 MHz, CDCl_3_): δ 7.18 (s, *H_ar_*, 4H), 4.30 (s, Cl-*CH_2_*-CO, 4H) ^13^C-NMR (300 MHz, CDCl_3_): δ 122.41 (C_ar_), 40.93 (Cl-*C*-CO) ESI^+^-MS: C_10_H_8_Cl_2_O_4_ [M + Na]^+^
*m*/*z*_calc_ 284.96919 *m*/*z*_found_ 284.96962.

#### 3.2.4. Synthesis of Tri-tert-butyl 2,2′,2′′-(10-(2-(4-((tert-butyldimethylsilyl)oxy)phenoxy)-2-oxoethyl)-1,4,7,10-tetraazacyclododecane-1,4,7-triyl)triacetate (**4**)

The tert-Butyl triester of DO3A (280 mg, 0.54 mmol) was dissolved in MeCN (4.5 mL) and NaHCO_3_ (91 g, 1.09 mmol) was added. Compound **2** (196 mg, 0.65 mmol) was dissolved in MeCN (0.9 mL) and added to the solution.

The reaction was followed with MALDI-TOF (matrix: dithranol) and the mixture was stirred at room temperature for 26 h under argon. After this time, the solution was diluted with H_2_O (10 mL) and extracted with DCM (5 mL × 3). The combined organic layers were washed with H_2_O (5 mL × 3) and brine. After drying over MgSO_4_, the solvent was evaporated and the crude product was purified by column chromatography (9.5–9: 0.5–1 DCM: MeOH) to yield product **4** as a colourless oil. (247 mg, 0.31 mmol, 58%). ^1^H-NMR (500 MHz, CDCl_3_): δ 6.93 (d, *H_ar_*, 2H), 6.81 (d, *H_ar_*, 2H), 4.17–2.62 (m-bs, *H_ring_*+ *N*-*CH_2_*-CO, 24H), 1.45 (s, O*t*-*Bu*, 27H), 0.98 (s, Si-C-*CH_3_*, 9H), 0.19 (s, Si-*CH_3_*, 6H) ^13^C-NMR (300 MHz, CDCl_3_): δ 122.17 (C_ar_), 120.75 (C_ar_), 56.26, 28.24, 25.80,−4.31 (Si-CH_3_) ESI^+^-MS: C_40_H_70_N_4_O_9_Si [M + H]^+^
*m*/*z*_calc_ 779.49848 *m*/*z*_found_ 779.50212.

#### 3.2.5. Synthesis of Hexa-tert-butyl 2,2′,2′′-(10-(1,4-phenylene) bis-1,4,7,10-tetraazacyclododecane-1,4,7-triyl)acetate (**5**)

The *tert*-Butyl triester of DO3A (645 mg, 1.25 mmol) was dissolved in MeCN (4 mL) and DIPEA (0.43 mL, 2.51 mmol) was added dropwise. Compound **2** (100 mg, 0.38 mmol) was dissolved in MeCN (2.5 mL) and added to the solution.

The reaction was followed with MALDI-TOF (matrix: dithranol) and the mixture was refluxed for 22 h under nitrogen. The solvent was evaporated and the yellow oil was dissolved in DCM and washed with H_2_O (10 mL × 3) and brine.

After drying over MgSO_4_, the solvent was evaporated to yield crude compound **5** as a yellow plastic. The residue was purified by column chromatography (9.5–9:0.5–1 DCM: MeOH) to yield **5** (242 mg, 0.20 mmol, 52%) as a yellow plastic. ^1^H-NMR (500 MHz, CDCl_3_): δ 7.12 (bs, *H_ar_*, 4H), 4.02–2.04 (bs, N-*CH_2_*-CO + *H_ring_*, 48H), 1.50–1.17 (bs, O*t-Bu*, 54H) ^13^C-NMR (300 MHz, CDCl_3_): δ 122.24 (C_ar_), 58.81 (C_ring_) 55.04 (C_ring_), 28.23 (C_tBut_), 28.06 (C_tBut_) ESI^+^-MS: C_62_H_106_N_8_O_16_ [M + H]^+^
*m*/*z*_calc_ 609.38634 *m*/*z*_found_ 609.34983 Melting point: 65.7 °C.

#### 3.2.6. Synthesis of Dibenzyl 1,4,7,10-tetraazacyclododecane-1,7-dicarboxylate (**6**)

Under an inert atmosphere, cyclen (1 g, 5.80 mmol) was dissolved in CHCl_3_ (52 mL) and CbzCl (1.69 mL, 11.90 mmol) was added dropwise, resulting in the formation of a white precipitate. The mixture was heated to 60 °C and stirred for 1 h and 40 min. The pH at the end of the reaction was confirmed to be acidic. The solvent was evaporated under reduced pressure to yield a white solid mixed with a yellow solid. The solid was washed over a filter with Et_2_O and the residue was taken up with 4M aq. NaOH and extracted 4 times with Et_2_O. The organic layers were washed with 1M aq. NaOH (×2) and then dried over Na_2_SO_4_. The solvent was evaporated under reduced pressure to yield compound **6** (1.47 g, 3.33 mmol, 57%) as a colourless oil. ^1^H-NMR (500 MHz, CDCl_3_): δ 7.36–7.28 (m, *H_ar_*, 10H), 5.15 (s, O-*CH_2_*-bz), 3.51–3.35 (m, *H_ring_*, 8H), 2.99–2.70 (m, *H_ring_*, 8H) ^13^C-NMR (300 MHz, CDCl_3_): δ 128.77 (C_ar_), 128.37 (C_ar_), 128.12 (C_ar_), 67.63 (O-*C*-bz), 51.07 (C_ring_), 50.66 (C_ring_), 50.09 (C_ring_), 49.52 (C_ring_), 48.89 (C_ring_), 48.60 (C_ring_) ESI^+^-MS: C_38_H_58_N_8_O_16_ [M + H]^+^
*m*/*z*_calc_ 441.24963 *m*/*z*_found_ 441.25100.

#### 3.2.7. Synthesis of Dibenzyl 4,10-bis(2-(tert-Butoxy)-2-Oxoethyl)-1,4,7,10-tetraazacyclododecane-1,7-dicarboxylate (**7**)

Compound **6** (1.47 g, 3.33 mmol) was dissolved in MeCN (24 mL). DIPEA (11.59 mL, 66.55 mmol) and *tert*-butyl bromoacetate (0.98 mL, 6.66 mmol) was added. The reaction mixture was refluxed for 30 min. The solvent was evaporated to yield an oil and solid mixture. The oil was taken up with Et_2_O and washed with H_2_O. After drying over Na_2_SO_4_, the solvent was evaporated under reduced pressure to yield a light yellow oil as pure compound **7** (2.21 g, 3.30 mmol, 99%). ^1^H-NMR (500 MHz, CDCl_3_): δ 7.37–7.28 (m, *H_ar_*, 10H), 5.12 (s, *CH_2_*, 4H), 3.52–3.15 (m, *CH_2_*, 12H), 2.87 (bs, *H_ring_*, 8H), 1.43 (s, O*t-Bu*, 18H) ^13^C-NMR (300 MHz, CDCl_3_): δ 170.70, 156.62, 137.02 (C_ar_),128.62 (C_ar_), 128.03 (C_ar_), 81.11, 67.14 (O-*C*-bz), 56.14 (CH_2_), 54.18 (CH_2_), 46.72 (C_ring_), 28.35 (C_tBut_) ESI^+^-MS: C_38_H_58_N_8_O_16_ [M + H]^+^
*m*/*z*_calc_ 669.38579 *m*/*z*_found_ 669.38769.

#### 3.2.8. Synthesis of Di-tert-butyl 2,2′-(1,4,7,10-tetraazacyclododecane-1,7-diyl)diacetate (**8**)

To a suspension of Pd/C 10% (442 mg, 20% wt) in *iso*-propanol (50 mL), a solution of compound **7** (2.21 g, 3.30 mmol) in *iso*-propanol (30 mL) was added. Ammonium formate (9.57 g, 152 mmol) was added and the mixture was refluxed for 15 min under constant µwave irradiation (150 W). The catalyst was filtered off through celite. The solvent was evaporated and the white solid residue was taken up in 4M aq. NaOH (50 mL) and extracted with CHCl_3_ (100 mL × 4). The combined organic layers were dried over Na_2_SO_4_ and the solvent evaporated to yield **8** (1.30 g, 98%) as a yellow viscous oil that became a white solid after a few days’ exposure to air. ^1^H-NMR (500 MHz, CDCl_3_): δ 3.31 (s, OC*CH_2_*, 4H), 2.80 (bs, *H_ring_*, 8H), 2.61 (t, *H_ring_*, 8H), 1.46 (s, O*t-Bu*, 18H) ^13^C-NMR (300 MHz, CDCl_3_): δ 171.10, 81.15, 57.35 (CH_2_), 52.18 (CH_2_), 45.87 (C_ring_), 28.36 (C_tBut_) ESI^+^-MS: C_20_H_40_N_4_O_4_ [M + H]^+^
*m*/*z*_calc_ 401.31223 *m*/*z*_found_ 401.31280.

#### 3.2.9. Synthesis of Di-tert-butyl 2,2′-(4,10-bis(2-(4-((tert-butyldimethylsilyl)oxy)phenoxy)-2-oxoethyl)-1,4,7,10-tetraazacyclododecane-1,7-diyl)diacetate (**9**)

Compound **8** (350 mg, 0.87 mmol) was dissolved in MeCN (4 mL) and DIPEA (1.52 mL, 8.74 mmol) was slowly added under a nitrogen atmosphere. After a few minutes, a solution of compound **2** (550 mg, 1.83 mmol) in MeCN (2.2 mL) was added and the mixture was refluxed for 2 h. After this time, the dark brown mixture was concentrated and the resulting brown oil was dissolved in EtOAc and washed three times with H_2_O. After drying over Na_2_SO_4_, the solvent was removed under reduced pressure to yield a brown oil. The residue was purified by column chromatography (9.25:0.75 DCM/MeOH) to yield compound **9** as a pale yellow oil (451 mg, 0.48 mmol, 55%). ^1^H-NMR (500 MHz, CDCl_3_): δ 6.96 (d, *H_ar_*, 4H), 6.75 (d, *H_ar_*, 4H), 3.71–2.17 (m-bs, *H_ring_*+ N-*CH_2_*-CO, 24H), 1.25 (s, O*t-Bu*, 18H), 0.98 (s, Si-C-*CH_3_*, 18H), 0.19 (s, Si-*CH_3_*, 12H) ^13^C-NMR (300 MHz, CDCl_3_): δ 173.74, 173.01, 153.50, 144.44, 122.13 (C_ar_), 120.38 (C_ar_), 82.27, 55.86, 55.11, 27.83, 25.78, −4.34 (Si-CH_3_) ESI^+^-MS: C_48_H_80_N_4_O_10_Si_2_ [M + H]^+^
*m*/*z*_calc_ 929.54857 *m*/*z*_found_ 929.55713.

#### 3.2.10. Synthesis of 2,2′,2′′-(10-(2-(4-Hydroxyphenoxy)-2-oxoethyl)-1,4,7,10-tetraazacyclododecane-1,4,7-Triyl)triacetic acid (**L**^1^)

Compound **4** (100 mg, 0.13 mmol) was dissolved in DCM (0.32 mL) and TFA (0.22 mL, 3.47 mmol) was slowly added to the mixture. The reaction was followed with MALDI-TOF (matrix: dithranol) until the starting material could no longer be observed. The clear solution was stirred at room temperature for 2 days, after which it was concentrated under reduced pressure. The resulting brown oil was dissolved in the minimum amount of MeOH and triturated multiple times with Et_2_O to yield compound **L^1^** (25 mg, 0.12 mmol, 39%) as a white powder. ^1^H-NMR (500 MHz, MeOD): δ 7.02 (d, *H_ar_*, 2H), 6.77 (d, *H_ar_*, 2H), 4.01 (s, N-*CH_2_*-CO, 2H), 3.81 (s, N-*CH_2_*-CO, 4H), 3.62–2.47 (m, N-*CH_2_*-CO + H_ring_, 8H), 3.19–3.11 (m, *H_ring_*, 8H) ^13^C-NMR (300 MHz, MeOD): δ 144.47 (C4), 123.43 (C_ar_), 116.52 (C_ar_), 57.58, 54.27, 52.51 ESI^+^-MS: C_22_H_32_N_4_O_9_ [M + H]^+^
*m*/*z*_calc_ 497.22421 *m*/*z*_found_ 497.22337 Melting point: 175.2 °C.

#### 3.2.11. Synthesis of 10-(1,4-Phenylene) bis-1,4,7,10-tetraazacyclododecane-1,4,7-Triyl)acetate (**L**^2^)

Compound **5** (200 mg, 0.16 mmol) was dissolved in DCM (0.3 mL) and TFA (0.37 mL, 5.74 mmol) was added dropwise. The clear solution was stirred at room temperature for two days. After this time the solvent was evaporated and the resulting brown oil was dissolved in the minimum amount of MeOH and triturated with Et_2_O to yield compound **L^2^** (47 mg, 0.05 mmol, 32%) as a white powder. ^1^H-NMR (500 MHz, D_2_O): δ 7.28–6.93 (m, *H_ar_*, 4H), 4.04–3.08 (m, N-*CH_2_*-CO + *H_ring_*, 48H) ^13^C-NMR (300 MHz, D_2_O): δ 122.67 (C_ar_), 51.37 (C_ring_), 48.01 (C_ring_) ESI^+^-MS: C_38_H_58_N_8_O_16_ [M + H]^+^
*m*/*z*_calc_ 442.20581 *m*/*z*_found_ 442.20612 Melting point: 216.7 °C.

#### 3.2.12. Synthesis of 2,2′-(4,10-Bis(2-(4-hydroxyphenoxy)-2-oxoethyl)-1,4,7,10-tetraazacyclododecane-1,7-diyl)diacetic Acid (**L**^3^)

Compound **9** (326 mg, 0.35 mmol) was dissolved in DCM (0.17 mL) and TFA (0.80 mL, 12.28 mmol) was slowly added to the mixture. The clear solution was stirred at room temperature for 3 days. After this time, it was concentrated under reduced pressure. The brown oil was dissolved in a minimum amount of MeOH and triturated multiple times with Et_2_O to yield compound **L^3^** (75 mg, 0.13 mmol, 36%) as an off-white powder. ^1^H-NMR (500 MHz, D_2_O): δ 7.00 (d, *H_ar_*, 4H), 6.87 (d, *H_ar_*,4H), 3.96 (s, N-*CH_2_*-CO, 4H), 3.91 (s, N-*CH_2_*-CO, 4H), 3.60 (s, H_ring_, 4H), 3.53 (s, H_ring_, 4H), 3.20 (s, H_ring_, 4H), 3.15 (s, H_ring_, 4H) ^13^C-NMR (300 MHz, D_2_O): δ 122.56 (C_ar_), 116.01 (C_ar_), 51.36, 47.91 ESI^+^-MS: C_22_H_32_N_4_O_9_ [M + H]^+^
*m*/*z*_calc_ 589.25042 *m*/*z*_found_ 589.24967 Melting point: 214.1 °C.

### 3.3. Synthesis of the Complexes

The ligand L^n^ (1 eq) was dissolved in MeOH and Ln(OTf)_3_ (**1 eq** for *n* = 1, 3; **2 eq** for *n* = 2) was added to the solution. The reaction mixture was stirred at 60 °C for 48–72 h. After this time, the solvent was evaporated and the residue was dissolved in a minimal amount of MeOH and precipitated with diethyl ether to yield a white powder. The complexes were dried by the freeze-dry method, stored in sealed vials under argon and dried again at the oil pump before every use.

#### 3.3.1. Eu.**L**^1^

^1^H-NMR (500 MHz, D_2_O): δ 33.37 (s, H_a-ax_), 32.99 (s, H_a-ax_), 26.63 (s, H_a-ax_), 24.73 (s, H_a-ax_), 11.84 (s, H_a-ax_), 7.58 (s), −0.50 (s), −0.83 (s), −1.17 (s), −3.54 (s), −5.40 (s), −6.45 (s), −6.85 (s), −7.92 (s), −8.97 (s), −9.57 (s), -10.48 (s), −12.40 (s), −14.14 (s), −14.45 (s), −15.67 (s), −16.71 (s), −18.98 (s). ESI^+^-MS: C_22_H_30_EuN_4_O_9_ [M-BQ + Li + Na]^+^
*m*/*z*_calc_: 581.0859; *m*/*z*_found_: 581.12683.

#### 3.3.2. Eu.**L^2^**

^1^H-NMR (500 MHz, D_2_O): δ 33.45 (s, H_a-ax_), 33.04 (s, H_a-ax_), 26.57 (s, H_a-ax_), 24.68 (s, H_a-ax_), 13.16 (s, H_a-ax_), −0.55 (s), −0.85 (s), −1.13 (s, H_a-eq_), −2.00 (s), −3.52 (s), −4.19 (s), −5.36 (s), −6.38 (s), −6.77 (s, H_b-eq_), −7.30 (s), −7.94 (s, H_b-ax_), −8.77 (s), −8.95 (s), −9.57 (s), −10.36 (s), −12.35 (s), −14.09 (s), −14.47 (s, H_c_), −15.72 (s, H_c’_), −16.72 (s), −18.99 (s). ESI^+^-MS: C_38_H_54_Eu_2_N_8_O_16_ [M-Eu.**L^1^** + 2H]^2+^
*m*/*z*_calc_: 553.09497; *m*/*z*_found_: 553.09433.

#### 3.3.3. Eu.**L^3^**

^1^H-NMR (500 MHz, D_2_O): δ 32.81 (s, H_a-ax_), 26.48 (s, H_a-ax_), 24.55 (s, H_a-ax_), 12.58 (s, H_a-ax_), −0.50 (s), −1.26 (s, H_a-eq_), −4.07 (s), −5.33 (s), −6.87 (s, H_b-eq_), −7.68 (s, H_b-ax_), −8.88 (s), −9.63 (s), −10.37 (s), −12.28 (s), −14.28 (s, H_c_), −15.28 (s, H_c’_), −16.71 (s), −18.93 (s). ESI^+^-MS: C_28_H_35_EuN_4_O_10_ [M-2BQ + Li + Na]^+^
*m*/*z*_calc_: 740.15600; *m*/*z*_found_: 581.0859.

#### 3.3.4. Tb.**L^1^**

^1^H-NMR (500 MHz, D_2_O): δ 158.10 (s), 141.43 (s), 82.28 (s), −15.97 (s), −21.11 (s), −27.08 (s), −45.67 (s), −65.83 (s), −89.49 (s), −94.24 (s), −98.91 (s), −114.70 (s), −155.75 (s), −184.30 (s), −307.81 (s), −334.80 (s). ESI^+^-MS: C_22_H_30_TbN_4_O_9_ [M-BQ + Li + Na]^+^
*m*/*z*_calc_: 589.09059; *m*/*z*_found_ 589.13109.

#### 3.3.5. Tb.**L^2^**

^1^H-NMR (500 MHz, D_2_O): δ 259.21 (s), 136.45 (s), 82.55 (s), −15.86 (s), −26.89 (s), −37.37 (s), −41.28 (s), −45.65 (s), −65.68 (s), −65.90 (s), −95.32 (s), −98.89 (s), −114.68 (s), −155.90 (s), −184.35 (s), −227.85 (s), −388.21 (s). ESI^+^-MS: C_38_H_54_Tb_2_N_8_O_16_ [M-Tb.**L^1^** + 2H]^2+^
*m*/*z*_calc_: 561.10046; *m*/*z*_found_: 561.09983.

#### 3.3.6. Tb.**L**^3^

^1^H-NMR (500 MHz, D_2_O): δ 257.88 (s), 157.74 (s), 141.11 (s), 135.48 (s), 82.26 (s), −15.81 (s), −27.23 (s), −37.31 (s), −41.46 (s), −45.59 (s), −65.80 (s), −95.02 (s), −98.57 (s), −114.58 (s), −155.58 (s), −184.08 (s). ESI^+^-MS: C_28_H_35_TbN_4_O_10_ [M-2BQ + 2H]^+^
*m*/*z*_calc_: 561.10046; *m*/*z*_found_ 561.09930.

#### 3.4. Oxidation Tests

In a typical experiment, hydroquinone or alpha-tocopherol were dissolved in the solvent (HEPES 0.1M pH 7.2 + 1% Triton X-100) to concentration 0.04 M in a small vial. When HEPES buffer was employed, hydroquinone was first dissolved in the smallest amount of EtOH possible and HEPES buffer with 1% Triton X-100 was added. If necessary, the mixture was sonicated in a water bath to favour solubilisation. The test compound (hydroquinone or α-tocopherol) was added to the mixture at room temperature and stirred for 5 min before withdrawal of a sample and dilution to 0.1 mM in a volumetric flask and transfer of 3 mL of sample in a cuvette. The absorption measurements were acquired scanning the wavelength range 800–200 nm.

The complex was dissolved in HEPES buffer 0.1 M at pH 7.2 to concentration 0.04 M. 75 µL of solution were withdrawn, transferred in a cuvette and brought to volume (3 mL) by addition of 2.925 mL of H_2_O to have a resulting 1mM concentration. HOCl (from HOCl 5% *w*/*v* up to 25 equivalents for Eu.**L^1^**, 38 equivalents for Eu.**L^2^**, 28 equivalents for Eu.**L^3^**, 20 equivalents for Tb.**L^1^**, 22 equivalents for Tb.**L^2^**, 28 equivalents for Tb.**L^3^**) was added to the mixture at room temperature and the cuvette was intermittently shaken for 5 min. After this time, the measurements were obtained.

## 4. Conclusions

Mono-and diesters of hydroquinone were synthesised and coupled to cyclen-based carboxylate ligands. Three different ligands **L^1^**, **L^2^**, and **L^3^** were prepared and used to form europium(III) and terbium(III) complexes: Ln.**L^1^**, Ln.**L^2^**, and Ln.**L^3^**. In total, six new complexes of europium(III) and terbium(III) were synthesized, characterised and tested for their activity towards the HOCl using a testbed developed in order to give reproducible results when exploring oxidation reactions. The testbed is based on a HEPES buffer at pH 7.2, with Triton-X added to ensure solubility of lipophilic systems.

The complexes Ln.**L^1^** and Ln.**L^2^** were found to behave as DOTA-monoamide complexes, despite being mono-esters. NMR and luminescence spectra showed that Ln.**L^1^** is more flexible than diester Ln.**L^2^**. Ln.**L^3^** is a DOTA-diester, and the added steric bulk resulted in a more rigid structure. All investigated complexes reacted selectively with HOCl. The reaction resulted in that the esters were hydrolysed, converting all complexes via one or two steps to [Ln.DOTA]^−^. The conversion could be followed using luminescence spectroscopy with a direct intensity-based or a ratiometric readout. Independent of the stoichiometry of the reaction, in exceed of 15 equivalents were needed to fully convert the probes to [Ln.DOTA]^−^. Thus, we conclude that incorporating the hydroquinone unit in a lanthanide complex reduces the reactivity towards reactive oxygen species. Additionally, the reactivity of the hydroquinone esters is dictated by the structure of the lanthanide complex, as exemplified by the activity of the di-ester in Ln.**L^2^** and mono-esters in Ln.**L^3^**. The next step is to incorporate an efficient antenna and a hydroquinone caging unit in a single lanthanide(III) complex.

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
