# Peer review of "HOCl Responsive Lanthanide Complexes Using Hydroquinone Caging Units"

_molecules, 2020, doi:10.3390/molecules25081959_

Round 1

Reviewer 1 Report

The authors describe a very interesting study that focuses on the use of cyclen scaffolds with appended hydroquinone derived arms and their subsequent reaction with hypochlorous acid. The full synthesis of the ligands and the europium and terbium complexes are described in good detail. Appropriate characterisation data is provided for all the complexes. 

Luminescence (time-gated and steady-state) measurements were collected to study the use of the synthesized complexes as oxidation probes. The europium, terbium and the pi-pi* transitions are all utilized in ascertaining the suitability of the probes prepared. NMR-based titrations using hypochlorous acid are detailed which highlights the change in molecular architecture brought about through the oxidation of the appended arm(s).

The paper is very well written, with only a few minor spelling mistakes throughout the manuscript (not sufficient at all in detracting from the science being discussed). The results are well documented and there is sufficient evidence to support the conclusions drawn.

I recommend publication without any need for changes to the manuscript. I think that the work will be of interest to the community that is engaged with this subject despite, as the authors point out, the lack of a chromophore in the ligand sphere to boost the intensity of the luminescence of the chelated metal ion. 

Author Response

We thank the reviewer and have proof read the manuscript.

Reviewer 2 Report

The authors report in this article on the synthesis of three novel DOTA like ligands and on their corresponding Eu(III) and Tb(III) complexes (Eu.L1, Eu.L2, Eu.L3, Tb.L1, Tb.L2 ad Tb.L3). The novel complexes have been characterized by means of NMR spectroscopy and MS spectrometry. The emission properties of the complexes have been studied as well.

The synthesized complexes react with HOCl, a typical hROS species, by forming quinone and the corresponding DOTA complex (Ln.L + HOCl = Ln.DOTA). The reaction, can be followed via titration experiments, by using emission or 1H-NMR spectra.

The manuscript is scientifically sound and readable in all its parts. The reported results, even though the sensing ability of the complexes towards RONS is not exceptional, can be interesting for a broad readership.

I have only a major concern regarding this work: Line 158: “The synthesized ligands and complexes were characterized using mass spectrometry and NMR.”

However, the characterization of the complexes is not reported in the Materials and Methods section. The authors must complete this section and better characterize the obtained complexes. In particular for complexes Eu.L1, Eu.L2, Eu.L3, Tb.L1, Tb.L2 and Tb.L3, considering also their paramagnetic nature, elemental analysis for the isolated compounds must be performed.

A final curiosity. Is it possible to use hydrogen peroxide instead of HOCl? This would be interesting.

Author Response

We thank the reviewer, and have now included the characterisation in the manuscript text. The data was previously only supplied as supporting information.

Regarding the elemental analysis, the compounds were to reactive for this experiment to be performed with any confidence. The reviewer will also note that the compounds degrade in the LCMS analysis.

We have included the data and a short paragraph on the reactivity of the probes towards hydrogen peroxide in the body of the text at line 249. We have included an addition figure showing the data.

Round 2

Reviewer 2 Report

The manuscript is much improved and most of the comments have been addressed.